# Green Credit and Total Factor Carbon Emission Performance—Evidence from Moderation-Based Mediating Effect Test

**DOI:** 10.3390/ijerph19116821

**Published:** 2022-06-02

**Authors:** Lingling Cao, Huawei Niu

**Affiliations:** 1School of Economics and Management, China University of Mining and Technology, Xuzhou 221116, China; lingling.cao@163.com; 2Business School, Suqian University, Suqian 223800, China

**Keywords:** total factor carbon emission performance, advancement of industrial structure, green technology innovation, mediating effect, China

## Abstract

To achieve China’s new development pattern and the “dual carbon” goals, it is necessary to boost emission reduction and high-quality economic development simultaneously. Green credit (GC), consisting of environmental regulation and economic leverage, has a profound impact on improving total factor carbon emission performance (TFCEP). By selecting the panel data of 30 provinces and municipalities in China from 2001 to 2020, this paper constructs a series of panel models to analyze the transmission path of GC to TFCEP. The results indicate that the relationship between GC and TFCEP showed an “inverted-U-shaped” relationship. This is mainly because “energy-saving and emission reduction” first appeared in the government planning outline in 2006, and transition-friendly enterprises successfully transformed with low-interest green credit, thereby effectively improving their TFCEP. However, as environmental regulations continue to increase and the scale of green credit continues to expand, the efficiency of green credit allocation and internal conflicts with other environmental regulation policies are also emerging. At the same time, the advancement of industrial structure and green technology innovation had a significant mediating effect between GC and TFCEP; government quality has a strong moderating effect on the second stage of the mediating process. When GC reaches a certain scale, it tends to restrain TFCEP more in central and western China than in eastern China. Therefore, it is of great significance to continuously increase the scale of GC, promote the advancement of clean energy industrial structure, and improve green technology innovation.

## 1. Introduction

The report of the 19th National Congress of the Communist Party of China pointed out that China’s economy has entered a critical period of transforming its growth impetus and optimizing its economic structure. Promoting green, low-carbon, and sustainable economic development has become a strategic direction for China’s social and economic development [1]. Mitigating climate change and promoting low-carbon economic transformation are the two significant challenges facing China and humanity in the 21st century. To meet the challenge and ensure that the global temperature rise is limited to 1.5 °C, China has made a solemn commitment to the international community to “strive to achieve carbon peaking by 2030 and carbon neutrality by 2060”. Undoubtedly, promoting low-carbon transformation and development is an inevitable choice to achieve the “dual-carbon” goals.

Both neoclassical and endogenous growth theories believe that economic growth driven by factor input and quality improvement by total factor productivity (TFP) tend to trade off each other [2,3]. Under the backdrop of global ecological governance, a consensus has been reached to incorporate environmental factors into the TFP framework to balance the sustainable and healthy development of the economy, and therefore, the concept of green total factor productivity (GTFP) was put forward at this moment [4]. Energy and environmental constraints are becoming increasingly tight. Compared with the single-factor carbon productivity indicator, incorporating carbon emissions and energy factors into the TFP growth measurement framework to measure total factor carbon emissions performance (TFCEP) is obviously more conducive to achieving the parallel goals of economic growth and emission reduction [5].

Coping with climate change and promoting low-carbon economic transformation and development requires considerable funds for renewable resources, energy efficiency improvement, and green infrastructure construction projects (OECD, 2017). Insufficient capital supply and inefficient capital allocation have become major constraints on the current low-carbon transformation [6]. Green finance has become a new engine to ease the above-mentioned constraints. In September 2015, the State Council issued the “Overall Plan for the Reform of Ecological Civilization System”, which formally proposed a green financial system and completed the top-level design of green financial development. As China is a typical bank-led country, green credit (GC) has become the core of the green financial system and will become the major means for green project financing [7]. In 2012, the “Green Credit Guidelines” was promulgated and implemented, which put forward clear requirements for the banking system to promote energy conservation, emission reduction, and environmental protection. Since then, local governments and financial institutions have responded positively, and the scale of green credit has constantly expanded. By the end of 2021, the green credit balance of China’s 21 major financial institutions has reached CNY 15.1 trillion, accounting for 10.6% of the country’s total loans, ranking first in the world. Meanwhile, the proportion of clean energy consumption in 2020 has reached 24.3% in China, an increase of 9.8 percentage points compared to 2012; the proportion of coal consumption has dropped to 56.8%, and the energy structure has been continuously optimized.

Green finance has played an important role in promoting low-carbon economic transition [8,9]. Consensus on the impact of green finance on TFP, however, has not been achieved [6,10,11]. Though the research on the relationship between finance and TFP is quite rich, few studies have examined the impact of GC on TFCEP. As a single environmental regulation measure, does green credit contribute to TFCEP? Is there a nonlinear relationship between them? How does the mediation mechanism affect their relationship? Does the government play a role in the process of green credit affecting TFCEP? For the above problems, there is still a lack of empirical analysis. To solve these problems, this paper constructs a series of panel models to analyze the transmission path of GC to TFCEP by selecting the panel data of 30 provinces and municipalities (due to the lack of sample data, the Tibet Autonomous Region, Hong Kong, the Macau Special Administrative Region, and Taiwan are not covered for the time being) in China from 2001 to 2020. The research conclusions of this paper have strong practical significance for improving the green credit system and achieving high-quality economic development. The major contributions of this paper are as follows: First, using one indicator—TFCEP—to represent two goals of emission reduction and economic growth can broaden the research perspective in analyzing the impact of GC on TFCEP. Second, a new data envelopment analysis (DEA) model based on a non-angular and non-radial directional distance function (DDF) is used to measure TFCEP more accurately, for it can avoid the problems of no solution in DDFs, biased angle, and radial measurement. Third, the mediating effect model is developed to outline the mechanism of GC on TFCEP, which confirms the mediating roles of green technology innovation (GTI) and advancement of industrial structure (AIS). Fourth, as a type of environmental regulation, carbon emission reduction measures should be regulated and guided by the government because they may not function well in a free market. This paper adopts a moderating effect model and finds that government quality (GQ) has moderating effects on two mediating chains.

## 2. Literature Review

### 2.1. The Concept and Measurement of TFCEP

Initially, scholars used a single factor such as energy intensity or carbon emissions per unit of energy to measure carbon emissions performance [12,13]. With the ubiquitous application of the concept of total factors, scholars have integrated input factors, energy consumption, and carbon emissions into performance evaluation, resulting in TFCEP [14]. TFCEP is defined as the ratio of potential carbon intensity to actual carbon intensity [15,16]. There are abundant research results on TFCEP measurement [17,18,19]. Since the nonparametric data envelopment analysis (DEA) model can effectively deal with the problems of irregular data and nonuniform dimensions [20], it has incomparable advantages compared to parametric analysis methods for efficiency measurement research that includes both expected and undesired outputs [16,21,22]. Based on the Shephard distance function, Chung et al. (1997) proposed the directional distance function (DDF) and constructed the Malmquist–Luenberger (ML) index under the assumption that the expected output and undesired output increase and decrease in the same proportion to calculate TFP growth considering environmental pollution emissions [3]. However, the ML index method’s—such as intertemporal mixed DDF—may have no solution, and the angular and radial efficiency measures are biased [22,23]. The Global ML index based on the frontier of global technology can solve DDFs’ no solution problem [24], but it does not solve the defects of angular and radial DEA methods. To solve this problem, non-angular, non-radial DDFs containing environmental undesired outputs have been developed [15,16,25,26]. The combination of global technology frontier and non-radial and nonangular DDFs can completely solve the problem of insoluble DDFs, but the convex combination of intertemporal observations may contain infeasible parts, including measurement bias. Based on the overall technical frontier DEA model proposed by Afsharian [27], Shao (2022) constructed a new DEA model of a non-angular and non-radial DDF based on global technology [28] which can exclude production infeasible areas.

### 2.2. Research on Financial Promotion of Emission Reduction and Improvement of TFP

Domestic and foreign research results on the influencing factors of carbon emission reduction are extremely rich, and it has been identified that industrial structure, financial development, urbanization and technological innovation are all key driving factors [15,19,29,30]. However, there is currently no consensus among academics on the impact of finance on carbon emission reduction. Some scholars suggested that by optimizing the allocation of resources [31], green finance could guide the influx of funds into environment-friendly sectors, optimize industrial structure, and promote carbon emission reduction and economic upgrading [32,33]. However, some scholars, based on data from different countries, found that finance has a certain inhibitory effect on carbon emission reduction [34,35]. Meanwhile, others proved that there is a U-shaped relationship between financial development to promote economic growth, increasing energy consumption, and increasing carbon emissions [36,37]. The main reason for the inconsistent conclusions is that the dimensions of financial development are different. The financial research on GTFP, which includes both desired and undesired outputs, has also formed a series of results, but the conclusions are not completely consistent [31,38,39]. Lee et al. (2022) believe that finance can significantly contribute to GTFP [39]. Some scholars also believe that green finance and GTFP had a nonlinear relationship and that a threshold exists [31,38].

In summary, there are many research results on TFCEP measurement, but the accuracy of the measurement results must be improved. At the same time, empirical research on green finance and TFCEP is still relatively rare. Is there a mediating effect between them? Does the government play a role in the process of green credit affecting TFCEP? These questions must be answered urgently.

## 3. Theoretical Analysis and Research Hypotheses

### 3.1. Direct Impact of GC on TFCEP

GC has an overall impact on the promotion of TFCEP by optimizing resource allocation and providing social supervision. First, GC optimizes the allocation of resources at both macro and micro levels to achieve the parallel goals of high-quality economic development and emission reduction. On one hand, under the “dual carbon” goals, micro-enterprises are forced to engage in clean production projects that will reduce business profits. For environmentally friendly enterprises, GC can reduce their financing costs and help improve business performance. However, for heavily polluting enterprises, GC forms financing constraints on them, such as reducing the supply of funds or greatly increasing their financing costs. The decline in financing capacity will directly lead to a decline in their production, and then indirectly lead to a reduction in the energy consumption and pollution emissions of the enterprises. On the other hand, traditionally high-pollution industries, such as the iron, steel, coal, and metallurgy industries, are mostly state-owned with monopoly status in China. Due to preferential policies, state-owned enterprises occupy a large amount of low-cost financing resources. As GC supports clean industries with high innovation ability, it forms financing constraints on traditional high-polluting state-owned enterprises, which pushes them to reduce carbon emissions [38]. Second, GC comprehensively manages loan entities from pre-loan approval to post-loan supervision. Only companies or projects that meet environmental monitoring standards and have effective pollution control measures can receive credit quotas. In order to prevent “greenwashing”, the regulatory authorities will supervise enterprises to ensure that their funds are invested in low-carbon technologies and environmental protection projects. Third, GC policies, rather than administration-mandated environmental regulations, are favored by local governments. In order to meet the assessment targets of energy conservation and environmental protection, local governments will provide administrative guidance or intervene in financial institutions to increase the scale of GC. At the same time, the effect of energy-saving and emission reduction will radiate to adjacent areas [32].

Green credit focuses on supporting technology-intensive or capital-intensive energy conservation and environmental protection industries. These industries often have a long investment cycle, which will increase investment and reduce output in the short term. In the long run, with the continuous expansion of the scale of green credit, the green technology level of enterprises will receive more financial support, which will enhance the enthusiasm of enterprises for green technology innovation and green production and drive the improvement of TFCEP.

**Hypothesis** **1.**
*There is a nonlinear relationship between GC and TFCEP.*


### 3.2. Indirect Effect of GC on TFCEP

#### 3.2.1. GC Improves TFCEP by Stimulating GTI

The Porter hypothesis points out that effective environmental regulation policies can stimulate technological innovation in enterprises [28]. As an environmental regulation policy, GC can force enterprises to make a trade-off between short-term financial performance and long-term low-carbon performance because it encourages the upgrading of traditional energy-intensive operations with GTI. Along with a detailed information disclosure policy, GC can promote enterprises to fulfill their social responsibilities; at the same time, differences in financing costs and constraints of GC promote high-pollution enterprises to adopt low-carbon technology. Therefore, GC can guide enterprises to pursue long-term low-carbon operation by improving their environmental protection awareness [40]. Changes in the external environment will passively force enterprises to choose GTI because high-energy-consumption and high-pollution enterprises must survive the competition [41].

The quality of government is the embodiment of the comprehensive management level of the government. Because the spontaneous adjustment of the market often lags behind and is unstable, GTI normally depends more on the government because the government controls resources and can intervene in resource allocation. Therefore, GQ is not only related to the effective implementation of GC policies, but also determines the effect of GC on improving TFCEP. Efficient government can guarantee that GC is used to promote energy conservation and emission reduction. First, effective government can monitor the access threshold of GC and improve the environmental quality of the whole society by increasing financial support in energy conservation and environmental protection [42]. Second, environmental protection subsidies have a significant “crowding out” effect on enterprises implementing green innovation, and sewage charges force them to carry out GTI [43]. Scholars such as Zhu et al. have verified that “effective government” can strengthen the restraining of GTI on CO_2_ emission intensity [44].

**Hypothesis** **2.**
*GC affects TFECP by upgrading GTI. The higher the quality of government, the more significant the positive relationship between GTI and TFCEP.*


#### 3.2.2. GC Improves TFCEP by AIS

GC can effectively promote the upgrading of industrial structure. The policies and low interest rates of GC will guide enterprises to transfer to clean and environmentally friendly industries [45] and increase the financing costs of high-energy-consumption and high-pollution enterprises. A strict information disclosure policy can effectively reduce the risk of information asymmetry, fully improve the allocation efficiency of financial resources, and promote the advancement of clean energy industrial structure [8]. The flow of GC funds can actively convey the value concept of green economy to the society, and GC policy can effectively guide start-ups to energy conservation and environmental protection industries and further optimize the regional industrial structure. A consensus of upgrading industrial structure to improve carbon emission performance has been reached [44]. The advancement of clean energy industrial structure can not only effectively reduce carbon emissions, but also generate high-end added value and drive high-quality economic development. Among the solutions, reducing the proportion of secondary industry is the best and most effective way to reduce carbon emissions [46].

Effective government strengthens the impact of AIS on TFCEP through two approaches. First, it can avoid negative externalities of environmental pollution which will lead to the failure of market mechanisms. Local governments adopt strict environmental regulations, such as carbon taxes and carbon quotas, to guide industrial enterprises to green transformation and industrial structure upgrading. Secondly, the current international political and economic environment is complex and changeable, and the negative impact of the global financial crisis and public health emergencies on the global economy is extremely severe. An effective government can hedge against external shocks through policy regulations. Since the outbreak of COVID-19 in 2020, the Chinese government has implemented combined measures in the fields of finance, currency, and employment which not only maintain the bottom line of national life, health and safety, but also guide industrial structure upgrading by vigorously developing the digital economy and leading the world economy in 2021.

**Hypothesis** **3.**
*GC affects TFECP by optimizing AIS. The higher the quality of government, the more significant the positive relationship between the AIS and TFCEP.*


The specific influence mechanism is shown in Figure 1.

## 4. Variable Selection and Model Building

### 4.1. Variable Selection

#### 4.1.1. Dependent Variable: Total Factor Carbon Emissions Performance (TFCEP) 

The overall production possibility set (PPS) under energy and environmental constraints is constructed as follows:(1)$$P^{0}(x_{t})=\overset{T}{\underset{t=1}{\cup }}P^{t}=\left\{\begin{array}{c} (y,b):(\sum_{k=1}^{K}z_{k}^{1}y_{km}^{1}\geq y_{km}^{t},\sum_{k=1}^{K}z_{k}^{1}b_{ki}^{1}=b_{ki}^{t},\sum_{k=1}^{K}z_{k}^{1}x_{kn}^{1}\leq x_{kn}^{t});or\cdots or\\ (\sum_{k=1}^{K}z_{k}^{T}y_{km}^{T}\geq y_{km}^{T},\sum_{k=1}^{K}z_{k}^{T}b_{ki}^{T}=b_{ki}^{T},\sum_{k=1}^{K}z_{k}^{T}x_{kn}^{T}\leq x_{kn}^{T});\\ z_{k}^{t}\geq 0,\forall m,\forall i,\forall k,\forall n, \end{array}\right\}$$
where, assuming the existence of *K* decision making units (DMUs), each DMU takes in N factors $x=(x_{1},\cdots x_{N})$ and produces M expected outputs $y=(y_{1},\cdots y_{M})$ and I undesirable outputs $b=(b_{1},\cdots b_{I})$. $\left(y_{k}^{t},b_{k}^{t},x_{k}^{t}\right)$ represents the input–output vector of ${\mathrm{DMU}}_{k}(k=1,\cdots K)$ and $z_{k}^{t}$ is the weight of cross-sectional observations in constructing the technological frontier.

The nonangular, nonradial DDF based on the global technique is expressed as:(2)$${\vec{D}}^{0}(x,y,b;g)=\sup\left\{\omega^{T}\beta :(x,y,b)+g\times diag(\beta )\in P^{0}(x_{t})\right\}$$
where $\omega ={\left(\omega_{m}^{y},\omega_{i}^{b},\omega_{n}^{x}\right)}^{T}$ represents the weight vector of input–output factors; $g=(g_{y};-g_{x};-g_{b})$ is the direction vector, which represents the expansion of expected output and the reduction of input and undesirable output; $\beta ={\left(\beta_{my},\beta_{nx},\beta_{ib}\right)}^{T}\geq 0$ is the proportion factor, which represents the expansion of expected output and the reduction of input and undesirable output. The larger the DDF value, the lower the input–output efficiency. When DDF = 0, it means that it is at the production frontier. Specifically, the DDF of period t can be solved by the following linear programming model: (3)$${\vec{D}}^{0}(x^{t},y^{t},b^{t};g^{t})=\max\left\{\left\{\begin{array}{c} \max\omega_{m}^{y}\beta_{my}^{0,t}+\omega_{i}^{b}\beta_{ib}^{0,t}+\omega_{n}^{x}\beta_{nx}^{0,t}\\ s.t.\sum_{k=1}^{K}z_{k}^{t}y_{km}^{t}\leq y_{m}^{t}+\beta_{my}^{0,t}g_{my}^{t},\forall m;\\ \sum_{k=1}^{K}z_{k}^{t}b_{ki}^{t}=b_{k}^{t}-\beta_{ib}^{0,t}g_{ib}^{t},\forall i;\\ \begin{array}{l} \sum_{k=1}^{K}z_{k}^{t}x_{kn}^{t}\geq x_{n}^{t}-\beta_{nx}^{0,t}g_{nx}^{t},\forall n;\\ z_{k}^{t}\geq 0 \end{array} \end{array}\right\},t=1,\cdots T\right\}$$

In this paper, we choose the deflator sequence of GDP (y) as the expected output, carbon dioxide emissions (b) as the undesirable output, capital stock (K), and labor employment (L) and energy consumption (E) as the input; the direction vector can be expressed as g=(y;−b;−K;−L;−E). According to the Luenberger productivity index, the TFCEP in period t+1 is:(4)$$\mathrm{TFCEP}={\vec{D}}^{0}(x^{t},y^{t},b^{t};g^{t})-{\vec{D}}^{0}(x^{t+1},y^{t+1},b^{t+1};g^{t+1})$$
when TFCEP > 0, it means TFCEP has improved.

#### 4.1.2. Explanatory Variable: Green Credit (GC)

GC data are normally collected to represent the social responsibility of different financial institutions or of the nation, but few are collected to represent that of provinces. Most of the existing literature uses indirect methods to measure GC at provincial level, mainly including bank loan balance in industrial pollution control investment, and proportion of loans for energy conservation, environmental protection projects, etc. In view of the completeness and availability of data, the indicators are positively processed, the scale of interest of industries excluding the six high-energy-consumption sectors is used to measure the level of GC, and logarithmic processing is carried out.

#### 4.1.3. Mediating Variable 

Advancement of industrial structure (AIS): AIS refers to the evolution process of an industry from a low-level state to an advanced state. In secondary industry, high-pollution and high-energy-consumption enterprises account for a relatively high proportion, while the energy consumption of tertiary industry is generally low. Therefore, promoting the development of tertiary industry has become an important means for various regions to practice energy conservation and emission reduction and alleviate the deterioration of link quality. Considering the practical problems of environmental resource constraints, many scholars use the proportion of added value of tertiary industry and secondary industry to measure the AIS [45,47].

Green technology innovation (GTI): Previous studies mainly used R&D investment or the number of patents to represent technological innovation but failed to highlight the “green” feature. The Green patent list issued by the World Intellectual Property Organization clearly defines the classification standard (IPC code) of green patents. Based on the green IPC code, we searched items by region and year, used the sum of green invention patents and green utility model patents to measure GTI, and processed the variables logarithmically.

#### 4.1.4. Moderating Variable: Government Quality (GQ)

We selected four subdimension indicators in the China Marketization Index compiled by Wang et al. (2021) [48], which include the reduction of corporate tax burden index, the government scale reduction index, the legal environment for maintaining the market index, and the intellectual property protection index. The mean value of the index reflects the level of GQ. The larger the value, the more effective the local government is.

#### 4.1.5. Control Variables

Four control variables—the degree of openness (OP), energy structure (ES), education level (EL), and environmental regulation (ER)—are taken into the regression equation. OP is obtained by performing logarithmic processing on the amount of foreign direct investment actually utilized; the proportion of coal consumption in each province and city against the total national coal consumption in that year is reflected by ES; and logarithmic processing results on the average years of education is represented by EL. ER, which is used to reflect the government’s emphasis on environmental protection, represents the proportion of words related to environmental protection in the provincial and municipal government reports.

Due to the lack of data from Hong Kong, Macau, Taiwan, and Tibet, only the data of the other 30 provinces and municipalities from 2001 to 2020 were collected for empirical analysis. The main sources include *China Statistical Yearbook*, *China Environment Yearbook*, *China Energy Statistical Yearbook*, *China Industrial Statistical Yearbook and Provincial Statistical Yearbooks*, *Marketization Index Report by Provinces in China* [48], local government work reports, and the website of the State Intellectual Property Office. The descriptive statistics and correlation analysis of all variables are shown in Table 1.

As shown in Table 1, the standard deviation coefficient of TFCEP is the largest, reaching 12.80, indicating a wide between-province gap in TFCEP, while the standard deviation coefficients of other variables are less than 1. The selected core variables have significant correlations, and there is no serious multicollinearity problem between variables, which ensures the feasibility of subsequent research.

### 4.2. Model Building

Considering that there are lag effects such as intertemporal correlation in TFCEP, and there exists a significant serial autocorrelation, the first-order lag of the dependent variable is introduced to construct a dynamic panel model:(5)$${\mathrm{TFCEP}}_{it}=\alpha_{0}+\alpha_{1}{\mathrm{TFCEP}}_{i,t-1}+\alpha_{2}{\mathrm{GC}}_{it}+\alpha_{3}{\mathrm{GC}}_{it}{}^{2}+\alpha_{4}Cont_{it}+\varepsilon_{it}^{1}$$

#### 4.2.1. Mediating Effect Model

Considering the impact mechanism, the mediating effect models based on AIS and GTI are constructed. However, since there is no significant serial autocorrelation between the mediating variables, the fixed-effect panel models are selected for GC and mediating variables, shown as follows:(6)$$\mathrm{AIS}=\beta_{0}+\beta_{1}{\mathrm{GC}}_{it}+\beta_{2}Cont_{it}+\varepsilon_{it}^{2}$$
(7)$${\mathrm{TFCEP}}_{it}=\gamma_{0}+\gamma_{1}{\mathrm{TFCEP}}_{i,t-1}+\gamma_{2}{\mathrm{GC}}_{it}+\gamma_{3}{\mathrm{GC}}_{it}{}^{2}+\gamma_{4}\mathrm{AIS}+\gamma_{5}Cont_{it}+\varepsilon_{it}^{3}$$
(8)$$\mathrm{GTI}=\lambda_{0}+\lambda_{1}{\mathrm{GC}}_{it}+\lambda_{2}Cont_{it}+\varepsilon_{it}^{4}$$
(9)$${\mathrm{TFCEP}}_{it}=\eta_{0}+\eta_{1}{\mathrm{TFCEP}}_{i,t-1}+\eta_{2}{\mathrm{GC}}_{it}+\eta_{3}{\mathrm{GC}}_{it}{}^{2}+\eta_{4}\mathrm{GTI}+\eta_{5}Cont_{it}+\varepsilon_{it}^{5}$$
where i represents the province, t the time, $Cont_{it}$ the control variable, and ε the residual. Specifically, after all variables are centralized, $\alpha_{1}$ represents the total effect of GC on TFCEP; in the mediating chain of AIS, $\gamma_{2}$ represents the direct effect of GC on TFCEP, and $\beta_{1}\times \gamma_{4}$ represents the indirect effect of GC on TFCEP; in the mediating chain of GTI, $\eta_{2}$ represents the direct effect of GC on TFCEP, and $\lambda_{1}\times \eta_{4}$ the indirect effect of GC on TFCEP.

#### 4.2.2. Moderation-Based Mediating Model

Referring to Zhu et al. [49], a moderation-based mediating model is constructed as follows: (10)$$\begin{array}{ll} {\mathrm{TFCEP}}_{it}= & a_{0}+a_{1}{\mathrm{TFCEP}}_{i,t-1}+a_{2}{\mathrm{GC}}_{it}+a_{3}{\mathrm{GC}}_{it}{}^{2}+a_{4}\mathrm{AIS}+a_{5}\mathrm{GQ}\\ & +a_{6}\mathrm{AIS}\times \mathrm{GQ}+a_{7}Cont_{it}+\varepsilon_{it}^{6} \end{array}$$
(11)$$\begin{array}{ll} {\mathrm{TFCEP}}_{it}= & b_{0}+b_{1}{\mathrm{TFCEP}}_{i,t-1}+b_{2}{\mathrm{GC}}_{it}+b_{3}{\mathrm{GC}}_{it}{}^{2}+b_{4}\mathrm{GTI}+b_{5}\mathrm{GQ}\\ & +b_{6}\mathrm{GTI}\times \mathrm{GQ}+b_{7}Cont_{it}+\varepsilon_{it}^{7} \end{array}$$
where the coefficients of AIS×GQ and GTI×GQ represent the moderating effects of GQ (GQ can moderate the effects of AIS or GTI on TFCEP), and the moderation-based mediating effects are $a_{4}+a_{6}\mathrm{GQ}$ and $b_{4}+b_{6}\mathrm{GQ}$, respectively.

## 5. Results and Analysis

### 5.1. GC and TFCEP

In order to avoid differences in the magnitude of variables, all the numerical variables were standardized. At the same time, in order to avoid the endogeneity problem between variables and between variables and residuals, this paper adopted a two-step SYM-GMM method for estimation. In order to verify whether GC could promote TFCEP and whether there was a nonlinear relationship between them, the regression results were presented by gradually adding control variables to show the impact of control variables on the regression results, shown in Table 2.

The results in Table 2 show that GC is highly significant at the 1% confidence level and GC2 is at the 5% confidence level. The coefficient of GC is significantly positive, and the coefficient of GC2 is significantly negative, which reveals the nonlinear relationship—or an inverted U shape—between GC and TFCEP. Moreover, all regression models have passed the Hansen test and tests for first- and second-order serial correlation, indicating that the instrumental variables selected by the two-step SYM-GMM method are reasonable and the estimation results robust. With the continuous expansion of the scale of GC, TFCEP increases first and then decreases around the inflection point. According to the two regression coefficients of GC in Model (5), the inflection point is inferred. When the logarithm of GC reaches 2.6204 (i.e., the scale of GC is CNY 1.374 billion), GC can effectively improve TFCEP. When the scale of GC crosses the inflection point, the increase of GC will restrain the optimization of TFCEP. Therefore, the nonlinear relationship of Hypothesis 1 is proved, but the actual manifestation is opposite to the theory.

The “energy-saving and emission reduction” policy first appeared in the government’s planning outline in 2006. Since then, local governments at all levels have successively introduced numerous supporting measures to guide enterprises’ low-carbon transformation. Easy-to-transform or green enterprises successfully achieve transformation and expansion with low-interest green credit, driving the optimization of TFCEP. Although environmental regulation has been continuously strengthened and the scale of green credit has continued to expand, the allocation efficiency of green credit and the internal contradictions with other environmental regulation policies continue to emerge, which are embodied in: (1) Although green finance has been proved to be able to promote carbon emission reduction and economic growth [8,32,33], GC may restrain further optimization of TFCEP because TFCEP reflects the hedging between expected and undesirable outputs with the same factor input, especially when the economic growth driven by GC and the carbon emission reduction stimulated by GC are not synchronized or are taking a reversed trend. (2) The essence of GC is capital lending, so the repayment ability of borrowers decides whether commercial banks lend or not. As a result, many enterprises with green innovation technology but limited scale are still unable to obtain GC, resulting in a relatively slow carbon reduction rate. Especially in China, state-owned enterprises account for a relatively high proportion of traditional high-energy-consuming enterprises. Constrained by environmental regulations, these high-energy-consumption enterprises will attempt high-risk green credit rent-seeking and “greenwashing” behaviors. Since *Southern Weekly* released the “Annual Corporate Greenwashing Behavior Ranking” in 2009, several companies have touched the legal bottom line every year. There is a large gap between China’s green credit standards and international standards, so greenwashing behavior is characterized by strong concealment and partial greenwashing. Similar to traditional credit, the disorderly expansion of green credit results in a mismatch of green credit [50], and the low allocation efficiency inhibits the further optimization of TFCEP. (3) In the green and low-carbon environmental protection industry, many companies have obtained GC support many times. When low-carbon technology reaches a certain level, the marginal effect of GC in promoting carbon emission reduction will continue to decrease. (4) Given the social responsibility supervision function of GC policy, commercial banks tend to provide credit support to enterprises that already have green and clean technologies. Although traditional high-energy-consumption enterprises cannot receive GC support, their direct financing constraints are relatively loose because most of them are listed companies. By the end of 2020, the carbon price in Europe was EUR 50 per ton, while the carbon price in Shanghai was only CNY 41 per ton, and it was only CNY 15 per ton in Fujian. In China’s carbon emission market, carbon allowances are sufficient and low-cost, so green credit regulations are ineffective for many companies.

### 5.2. Test of Mediating Effect

Models (6)–(10) of Table 3 report the regression results of the mediating effect of AIS and GTI between GC and TFCEP. In Models (6) and (7), the coefficient of GC is highly significant at the 1% confidence level, indicating that GC can effectively promote AIS and stimulate GTI. The results of Models (8) to (10) show that the inverted-U-shaped relationship between GC and TFCEP is still significant after the mediating variables are added. When one or two mediating variables are put into the equation at the same time, the coefficient of the mediator variable is significant at the 10% confidence level; moreover, the regression coefficient of GC is lower than that of Model (5). This shows that with the continuous expansion of GC, the regional industrial structure will be upgraded and the GTI significantly enhanced. This, in turn, will drive the continuous optimization of TFCEP, which proves Hypotheses 2 and 3. From 2001 to 2020, the added value of secondary industry accounted for about 40% of the total economic output in 30 regions in China, but industrial carbon emissions accounted for more than 65% of the total emissions. Under the emission reduction targets, reducing the proportion of secondary industry could effectively improve TFCEP.

In order to further confirm the validity of the mediating effect, the bootstrap under 95% confidence interval was chosen to test the two mediating mechanisms again, and 2000 was selected as the sampling number. It can be seen from Table 4 that the two mediating chains show partial mediating effects, and neither the 95% percent confidence interval nor the bias-corrected confidence interval contains 0, which confirms the first half of Hypotheses 2 and 3. Comparing the two mediating mechanisms, we find that the mediating effect of GTI is stronger than that of AIS.

### 5.3. Moderating Effect Analysis

As mentioned above, in order to verify the moderating effect of GQ, the interaction between GQ and GTI or AIS is introduced into the regression model. The specific regression results and moderating effect diagrams are shown in Table 5 and Figure 1.

As shown in Table 5, the regression coefficient between TFCEP and the AIS–GQ interaction is 0.020 (*p* < 5%), indicating that GQ plays a positive moderating role between AIS and TFCEP; in the same way, the regression coefficient between TFCEP and the GTI–GQ interaction is 0.037 (*p* < 1%), indicating that GQ also plays a positive moderating role between GTI and TFCEP and that GQ has a stronger effect on the adjustment of GTI on TFCEP than AIS on TFCEP. As shown in Figure 2, the higher the GQ, the more effective the government is, and the more positive the correlation between AIS (or GTI) and TFCEP, which proves the second half of Hypotheses 2 and 3.

The mean value of GQ plus or minus one standard deviation is used for grouping, and the differences in the mediation effect under different groups are compared, as shown in Table 6. When GC affects TFCEP, neither the 95% confidence interval nor the bias-corrected confidence interval contains 0, and the mediating effect of GTI is significantly positive at different levels of GQ. Furthermore, the coefficient increased from 0.0815 for low GQ to 0.1375 for high GQ, which confirms Hypothesis 2—that the moderating effect of GQ on the mediating chain of GTI is significant. The mediating effect of AIS varies with the level of GQ. The mediating effect is significant in the average and high GQ group, but not in the low GQ group, which confirms Hypothesis 3—that the moderating effect of GQ on the mediating chain of AIS is significant.

### 5.4. Heterogeneity Analysis

Considering the vast differences in economic structure, development level, and government efficiency in different regions in China, the positive effect of GC on TFCEP may also vary greatly. The article refers to the division standard of the eastern, central, and western regions by the National Bureau of Statistics and analyzes the regional heterogeneity of the developed eastern regions (DER) and the underdeveloped central and western regions (CWR). See Table 7 for details.

The regression results in Table 7 show that the inverted-U-shaped relationship between GC and TFCEP is confirmed. Models (13) and (18) reveal that the inflection points of the inverted U shape are quite different between DER (4.2283) and CWR (2.1642), and the slope in DER is flatter than that in CWR. This means when the GC reaches a certain scale, CWR has a more obvious restraining effect on TFCEP than DER do. The reason is that (1) in the DER, GC scale is relatively large, and the incentives for GTI are also at the leading level in the country, so the DER have achieved a better balance between emission reduction and economic growth. With more environmental regulations issued and implemented in various regions of China, traditional high-energy-consumption enterprises have migrated from DER to CWR. This migration has, in CWR, accumulated a great many highly polluting enterprises that lag behind the DER in AIS and leave extremely severe pressure on emission reduction for these regions.

The effect of GC in the promotion of GTI in CWR and DER varies. Similarly, the mediating effect of GTI on the chain from GC to TFCEP also varies. Models (14) and (19) show that the promoting effects of GC on GTI are significant in all regions. However, the regression coefficient in CWR is significantly higher than that in DER due to the low level of GTI in CWR. Models (16) and (21) reveal that the regression results of GTI on TFCEP are significant, which confirms the existence of the mediating effect of GTI; however, the regression coefficient in DER is significantly higher than that in CWR, indicating that DER have better efficiency in achieving emission reduction and economic growth with GTI.

The mediating role of AIS in the three regions is also different. Models (15) and (20) indicate that GC promotes AIS, but the promotion effect, expressed in the regression coefficient, is more significant in DER than in CWR. Models (17) and (22) further prove that the regression coefficient of AIS on TFCEP is significant at the 10% confidence level in DER but is not significant in CWR. This means that in CWR, the mediating effect of AIS on the impact of GC on TFCEP does not exist due to the high proportion of secondary industry in CWR.

### 5.5. Robustness

In the benchmark regression model, in order to avoid the two-way causal relationship between GC and TFECP, this paper constructs a dynamic panel model and uses the two-step SYM-GMM method to estimate, and the results prove to be effective. In order to further verify the robustness of the results, outliers are removed, the length of time reduced, and municipalities excluded. The specific results are shown in Table 8.

(1)Removing outliers. Outliers may contaminate regression results, so 1% of dependent variables from two tails are removed.(2)Excluding municipalities. Compared with other provinces, the four municipalities directly under the Central Government (Beijing, Shanghai, Chongqing, and Tianjin) in China have significant advantages in terms of policies, location and transportation, historical and cultural gathering, and environmental governance. This may make the regression results more significant. Four municipalities are excluded for their unusual scale of economy and only the panel data of 26 provinces are retained for multiregression. Table 8 shows that the empirical results of GC on TFCEP are not significantly different from the previous regression results, which proves the robustness of the results.

## 6. Conclusions

To achieve China’s new development pattern and the “dual carbon” goals, it is necessary to boost emission reduction and high-quality economic development simultaneously. GC, consisting of environmental regulation and economic leverage, has a profound impact on improving TFCEP. By selecting the panel data of 30 provinces and municipalities in China from 2001 to 2020, this paper constructs a series of panel models to analyze the transmission path of GC to TFCEP. The specific conclusions are as follows:(1)There is an inverted-U-shaped relationship between GC and TFCEP. In the early stage, it was not difficult for green enterprises to successfully achieve transformation and expansion with low-interest green credit, which led to the optimization of TFCEP. However, under the situation in which environmental regulation is increasing and the scale of green credit is expanding, the mismatch of green credit is serious. The internal contradiction between green credit and other environmental regulation policies is also constantly manifested, which inhibits the improvement of total factor carbon emission performance.(2)GC improves TFCEP through AIS and GTI. Specifically, GC promotes AIS by improving the allocation efficiency of financial resources. Through financing constraints or incentives, it promotes enterprises to conduct low-carbon technology research and development, thereby comprehensively optimizing TFCEP. Moreover, because emission-reduction measures, as a type of environmental regulation, have little pressure on enterprises in the free market, the government should regulate and guide them in doing so. The results confirm that GQ plays a moderating role in the second stage of the two-stage mediating chains.(3)Heterogeneity analysis reveals that the inflection point of the inverted U shape in ER is located to the right of that in CWR, and the slope is also gentler. This means that when GC reaches a certain scale, it has stronger restraint on TFCEP in CWR than in ER. Due to the aggregation of the secondary industry in CWR, the mediating effect of AIS in this region is no longer significant.

## 7. Recommendations

Based on this study, the following policy implications can be drawn:(1)It is urgent to optimize the efficiency of green credit allocation. At present, although Chinese GC scale ranks as the top in the world, it only accounts for about 10% of all loans. There is still not enough funding for low-carbon technology upgrading in secondary industry. Commercial banks have insufficient drive to expand GC business, so it is necessary to establish a GC-risk-sharing mechanism that integrates government, commercial banks, policy banks, insurance, guarantees, and social capital. The government provides financial assistance to green projects and tax incentives to GC proceeds. For example, tax- and fee-reduction policies can aid industries undertaking low-carbon transitions. On the premise that policy banks increase GC, professional financial green policy institutions can be established to allocate green funds more accurately and efficiently. Insurance and guarantee institutions should be able to diversify and disperse GC risks. Relying on government reputation and subsidies, social capital can also be leveraged to directly participate in GC business. It is also necessary to strengthen government intervention to eliminate “greenwashing” with stricter environmental regulations, supervise the “fairness” of green credit, and ensure that more low-carbon green private enterprises can obtain low-interest loans.(2)Cross-department coordination can boost AIS. The government should improve the exit mechanism for high-energy-consumption and high-pollution enterprises—and especially avoid the westward migration of these enterprises—and accelerate the elimination of production sectors with low-efficiency and high-energy consumption. Quotas in the national emission trading market should be tighten, and carbon prices raised, so as to force high-emitting enterprises to improve energy efficiency. The government can reduce the direct financing constraints of low-carbon sectors by increasing the proportion of low-carbon small- and medium-sized enterprises listed on the New Third Board or Fourth Board. For the central and western regions, it is necessary to strengthen the government’s supervision and guidance, improve the environmental access threshold, and optimize industrial structure.(3)The government should take an active role in improving GTI. A package of government policies should be developed to promote GTI, such as increasing the government’s green purchasing efforts, setting up a special fund for low-carbon innovation, engaging the government into the application of new green and low-carbon technologies, providing an innovative technology platform for the deep integration of production, education and research, designing the layout of green and low-carbon industries in the region, etc. With these policies, China can effectively stimulate the market to participate in green innovation and improve the GTI level.

## Figures and Tables

**Figure 1 ijerph-19-06821-f001:**
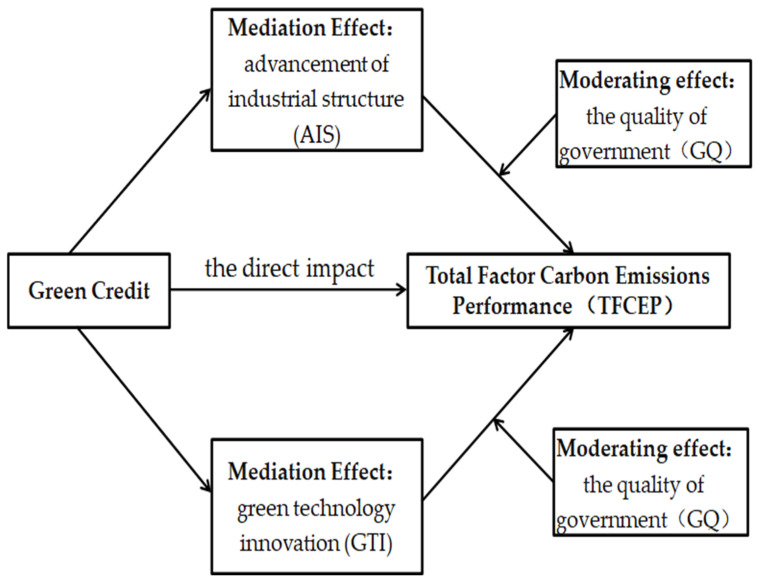
Diagram of the influence mechanism of GC and TFCEP.

**Figure 2 ijerph-19-06821-f002:**
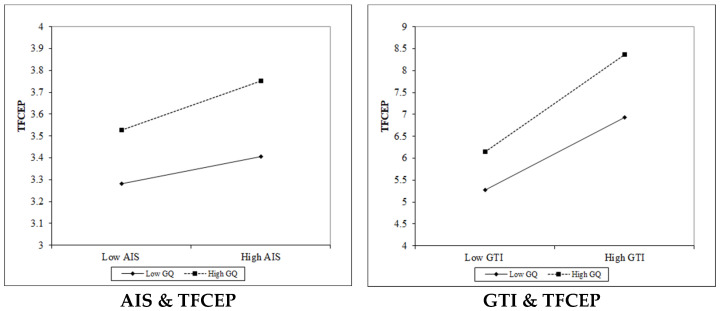
The moderating effect of government quality.

**Table 1 ijerph-19-06821-t001:** Descriptive statistics and correlation analysis of variables.

	Mean	Std	TFCEP	GC	AIS	gti	gq	op	es	el	er
TFCEP	0.178	2.279	1								
GC	7.120	0.586	0.383 ***	1							
AIS	1.171	0.624	0.268 ***	0.518 ***	1						
gti	6.974	1.939	0.411 ***	0.782 ***	0.233 ***	1					
gq	6.419	1.999	0.302 ***	0.660 ***	0.213 ***	0.702 ***	1				
op	12.061	2.185	0.128 ***	0.463 ***	0.019	0.551 ***	0.438 ***	1			
es	0.0333	0.0234	−0.012	0.251 ***	−0.277 ***	0.440 ***	0.453 ***	0.434 ***	1		
el	2.165	0.116	0.401 ***	0.762 ***	0.533 ***	0.684 ***	0.541 ***	0.261 ***	0.048	1	
er	0.0055	0.0026	0.208 ***	0.439 ***	0.126 ***	0.409 ***	0.128 ***	0.254 ***	0.017	0.255 ***	1

Note: *** *p* < 0.01 represents significant confidence levels of 1%.

**Table 2 ijerph-19-06821-t002:** SYM-GMM estimation results of green credit and TFCEP.

Variables	TFCEP					
	(0)	(1)	(2)	(3)	(4)	(5)
L.TFCEP	0.351 *** (8.62)	0.415 *** (10.79)	0.399 *** (10.32)	0.395 *** (10.23)	0.395 *** (10.21)	0.398 *** (10.25)
GC	0.442 *** (7.93)	0.332 *** (7.85)	0.371 *** (8.40)	0.307 *** (5.14)	0.310 *** (5.01)	0.283 *** (4.11)
GC2		−0.047 ** (−1.99)	−0.055 ** (−2.31)	−0.060 ** (−2.49)	−0.060 ** (−2.49)	−0.054 ** (−2.16)
ES			−0.103 *** (−2.88)	−0.090 ** (−2.47)	−0.087 ** (−2.24)	−0.080 ** (−2.01)
EL				0.087 * (1.78)	0.087 * (1.77)	0.095 * (1.70)
OP					−0.009 (−0.21)	−0.012 (−0.29)
ER						0.038 ** (1.94)
Constant	−0.001 (0.996)	0.056 (1.35)	0.060 (1.48)	0.063 (1.54)	0.064 (1.55)	0.057 (1.37)
province	yes	yes	yes	yes	yes	yes
year	yes	yes	yes	yes	yes	yes
R-squared	0.282	0.355	0.364	0.367	0.367	0.368
AR(1)	−2.68 (0.007)	−2.28 (0.022)	−2.33 (0.020)	−2.42 (0.015)	−2.29 (0.022)	−2.08 (0.037)
AR(2)	1.20 (0.229)	0.30 (0.767)	0.25 (0.800)	0.20 (0.841)	0.31 (0.759)	0.27 (0.791)
Hansen	28.05 (1.000)	26.64 (1.000)	28.09 (1.000)	24.58 (1.000)	24.64 (1.000)	22.49 (1.000)
N	570	570	570	570	570	570

Note: *** *p* < 0.01, ** *p* < 0.05, * *p* < 0.1, represent significant confidence levels of 1%, 5% and 10%, respectively, L.TFCEP is the first-order lag term of TFCEP.

**Table 3 ijerph-19-06821-t003:** Regression analysis of mediation mechanism.

Variables	AIS	GTI	TFCEP		
	(6)	(7)	(8)	(9)	(10)
L.TFCEP			0.398 *** (10.24)	0.381 *** (9.77)	0.380 *** (9.73)
AIS			0.022 * (1.73)		0.039 ** (2.58)
GTI				0.249 *** (3.11)	0.263 *** (3.14)
GC	0.760 *** (15.01)	0.476 *** (13.80)	0.274 *** (3.87)	0.155 * (1.94)	0.128 ** (2.40)
GC2			−0.047 ** (−2.48)	−0.036 ** (−2.43)	−0.047 ** (−2.51)
OP	−0.098 *** (−2.95)	0.141 *** (6.23)	−0.013 (−0.33)	−0.045 ** (−1.96)	−0.044 * (−1.75)
ES	−0.424 *** (−13.29)	0.245 *** (11.28)	−0.089 * (−1.89)	−0.139 *** (−3.17)	−0.127 *** (−2.64)
EL	0.047 (1.07)	0.248 *** (8.18)	0.096 * (1.71)	0.040 (0.68)	0.035 (0.60)
ER	−0.187 *** (−5.91)	0.097 *** (4.48)	0.037 ** (1.91)	0.022 (0.56)	0.023 ** (2.58)
Constant	−0.000 (−0.00)	0.000 (0.00)	0.050 (1.08)	0.032 (0.77)	0.043 (0.94)
province	yes	yes	yes	yes	yes
year	yes	yes	yes	yes	yes
R-squared	0.547	0.789	0.368	0.379	0.379
AR(1)			−2.45 (0.014)	−2.46 (0.014)	−2.26 (0.024)
AR(2)			1.20 (0.232)	1.15 (0.249)	1.06 (0.287)
Hansen			21.81 (1.000)	27.49 (1.000)	25.48 (1.000)
N	600	600	570	570	570

Note: *** *p* < 0.01, ** *p* < 0.05, * *p* < 0.1, represent significant confidence levels of 1%, 5% and 10%, respectively.

**Table 4 ijerph-19-06821-t004:** Test results of the mediating effect of bootstrap method.

Mediating Variable		Observed Coef.	Bootstrap Std. Err.	z	P [95% Conf. Interval]	BC [95% Conf. Interval]
GTI	indirect effect	0.1521	0.0352	4.33 ***	[0.0864, 0.2255]	[0.0876, 0.2272]
	direct effect	0.2561	0.0747	3.43 ***	[0.1055, 0.4062]	[0.1093, 0.4093]
AIS	indirect effect	0.0751	0.0312	2.41 ***	[0.0129, 0.1345]	[0.1374, 0.1512]
	direct effect	0.3331	0.0013	6.67 ***	[0.3406, 0.6254]	[0.3374, 0.6200]

Note: *** *p* < 0.01 represents significant confidence levels of 1%; (P): percentile confidence interval; (BC): bias-corrected confidence interval.

**Table 5 ijerph-19-06821-t005:** Regression results of moderated mediation effect.

Variables	(11)	(12)
	TFCEP	
L.TFCEP	0.396 *** (10.17)	0.380 *** (9.72)
GC	0.254 *** (2.91)	0.116 ** (2.36)
GC2	−0.043 ** (−2.03)	−0.051 * (−1.66)
AIS	0.012 * (1.75)	
GQ	0.051 * (1.89)	0.032 * (1.59)
GTI		0.261 *** (3.08)
AIS × GQ	0.020 ** (2.39)	
GTI × GQ		0.037 *** (2.78)
OP	−0.016 (−0.39)	−0.040 (−0.92)
ES	−0.100 ** (−2.06)	−0.155 *** (−3.31)
EL	0.097 * (1.67)	0.050 (0.83)
ER	0.046 (1.08)	0.023 (0.53)
Constant	0.049 (1.00)	0.021 (0.47)
R-squared	0.369	0.380
AR(1)	−2.43 (0.015)	−2.26 (0.024)
AR(2)	1.19 (0.233)	0.96 (0.337)
Hansen	21.46 (1.000)	20.73 (1.000)
N	570	570

Note: *** *p* < 0.01, ** *p* < 0.05, * *p* < 0.1, represent significant confidence levels of 1%, 5% and 10%, respectively.

**Table 6 ijerph-19-06821-t006:** Bootstrap test for the mediating effect of moderation.

Mediating Variables	GQ	Observed Coef.	Bootstrap Std. Err.	z	[95% Conf. Interval]	Significant or Not
GTI	−standard deviation	0.0815	0.0463	2.97 ***	[0.0467, 0.2299] P [0.0583, 0.2353] BC	yes
	mean	0.1095	0.0555	1.97 **	[0.0070, 0.2257] P [0.0120, 0.2311] BC	yes
	+standard deviation	0.1375	0.0747	1.69 *	[0.0526, 0.2390] P [0.05157, 0.2394] BC	yes
AIS	−standard deviation	0.00967	0.0209	1.32	[−0.0293, 0.0498] P [−0.0340, 0.0469] BC	no
	mean	0.03539	0.0268944	1.86 **	[0.0162, 0.0887] P [0.0124, 0.0953] BC	yes
	+standard deviation	0.0611	0.0413275	2.39 ***	[0.0206, 0.1379] P [0.0083, 0.1596] BC	yes

Note: *** *p* < 0.01, ** *p* < 0.05, * *p* < 0.1, represent significant confidence levels of 1%, 5% and 10%, respectively.

**Table 7 ijerph-19-06821-t007:** Summary of regional heterogeneity analysis.

Variables	DER					CWR				
	TFCEP (13)	GTI (14)	AIS (15)	TFCEP (16)	TFCEP (17)	TFCEP (18)	GTI (19)	AIS (20)	TFCEP (21)	TFCEP (22)
L.TFCEP	0.321 *** (4.96)			0.321 *** (4.94)	0.305 *** (4.65)	0.423 *** (8.68)			0.418 *** (8.57)	0.421 *** (8.64)
GTI				0.119 * (1.80)					0.111 * (1.76)	
AIS					0.088 * (1.82)					0.094 (0.87)
GC	0.389 *** (3.95)	0.359 *** (6.04)	0.820 *** (6.33)	0.385 *** (3.04)	0.380 *** (3.87)	0.549 *** (6.42)	0.827 *** (20.00)	0.164 *** (5.89)	0.435 *** (3.34)	0.525 *** (5.84)
GC2	−0.046 * (−1.72)			−0.045 ** (−2.10)	−0.086 * (−1.73)	−0.127 * (−1.74)			−0.125 * (−1.71)	−0.115 (−1.55)
Constant	−0.085 * (−1.79)	−0.299 *** (−7.81)	−0.078 * (−1.94)	−0.085 * (−1.78)	−0.045 (−1.42)	0.083 (1.51)	0.212 *** (7.61)	−0.128 *** (−5.50)	0.065 (1.14)	0.101 * (1.71)
Control	yes	yes	yes	yes	yes	yes	yes	yes	yes	yes
province	yes	yes	yes	yes	yes	yes	yes	yes	yes	yes
year	yes	yes	yes	yes	yes	yes	yes	yes	yes	yes
R-squared	0.278	0.825	0.676	0.274	0.281	0.402	0.832	0.082	0.397	0.396
N	209	220	220	209	209	361	380	380	361	361

Note: *** *p* < 0.01, ** *p* < 0.05, * *p* < 0.1, represent significant confidence levels of 1%, 5% and 10%, respectively.

**Table 8 ijerph-19-06821-t008:** Summary of various robust regressions.

Variables	1% Reduction	Exclude Municipalities
	TFCEP	TFCEP
L.TFCEP	0.423 *** (10.98)	0.429 *** (10.37)
GC	0.278 *** (4.27)	0.301 *** (6.45)
GC2	−0.051 ** (−2.17)	−0.048 * (−1.82)
Constant	0.058 (1.49)	0.036 * (1.75)
Control	yes	yes
province	yes	yes
year	yes	yes
R-squared	0.392	0.330
Observations	570	494

Note: *** *p* < 0.01, ** *p* < 0.05, * *p* < 0.1, represent significant confidence levels of 1%, 5% and 10%, respectively.

## Data Availability

The data presented in this study are available on request from the corresponding author. The data are not publicly available due to privacy.

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
