# Peer review of "Green Credit and Total Factor Carbon Emission Performance—Evidence from Moderation-Based Mediating Effect Test"

_ijerph, 2022, doi:10.3390/ijerph19116821_

Round 1

Reviewer 1 Report

The objective of the article is to analyze the transmission path of green credit (GC) to total factor carbon emission performance (TFCEP).

It lacks a better motivation for the paper. What kind of evidence/hypothesis the do authors want to show? Is a provincial panel used, and what are the relevant geographic patterns? Are they heterogeneous in terms of access to green credit? Is factor productivity heterogeneous? What is the sectoral role in the analysis? A better description of the green credit system in the analyzed country is lacking.

Section 2.

Many hypotheses are not clearly explained. Therefore, I suggest reducing core hypotheses, which should be associated with the regional/temporal case study.

H1: lack of a better link with current literature. It lacks a better relationship with the Chinese context.

H2. There is a lack of better linkage with the company's strategic decisions.

H3. Please define properly the advancement of industrial structure (AIS)

H4. There is no evidence to support the hypothesis.

H5. Please define clearly the quality of government and the expected cause-and-effect channels.

Methods.

There is potential endogeneity in the proposed model. Please discuss robustness strategies to deal with the problem. The strategy currently only masks the data – excluding outliers or spatial units – and does not show an alternative use of variables that might reduce the endogeneity problem. Please better justify the strategy and validity of the estimations.

Author Response

Thank you very much for your constructive comments for the improvement of our paper.I have responded item by item to your questions and suggested revisions.The specific information is in the Word text.

Reviewer 2 Report

Dear Authors

The topic is good, and the content is rich yet to improve

1- need to have a literature review section after the introduction

2- point out the gaps and answer the why question and why this study

3- conclusions and recommendations can be separate and be presented in  a better format 

4- After the data a reader expects results or findings section so please prepare it

5- Introduction is not satisfactory you need to contribute more and extended more

6- reference list needs to be more and you  need to consider other authors not just few

7- Review the keywords, such as add China

Author Response

(The authors gave the same response as above.)

Reviewer 3 Report

The concept of the paper has been well thought out. The title is consistent with the content of the article. The authors chose current and very important topic.

Nevertheless, the points indicated below should be clarified.

1. In 17 - 18 lines and in lines 321 and next we can find:

(17-18) "However, as environmental regulations continue to increase and the scale 17 of green credit continues to expand, the efficiency of green credit allocation and internal conflicts 18 with other environmental regulation policies are also emerging."

(321 and next) "Although 321 environmental regulation has been continuously strengthened and the scale of green 322 credit has continued to expand, the allocation efficiency of green credit and the internal 323 contradictions with other environmental regulation policies continue to emerge (…)"

My question is: What kind of internal conflicts are you talking about? This should be clarified. Maybe it's not about internal conflicts, but about an increase in the detection of greenwashing cases?

Environmental Policy should be coherent. If it is inconsistent, it proves that the government's actions are not effective enough.

2. I have comments on the validity of some hypotheses.

You have such hypothesis:

Hypothesis 1: GC has a positive effect on TFCEP. 107 

Hypothesis 2: GTI has a mediating effect between GC and TFCEP. 122 

Hypothesis 3: The AIS has a mediating effect between GC and TFCEP. 138 

Hypothesis 4: The higher the GQ, the more significant the positive relationship be- 152 tween GTI and TFCEP. 

Hypothesis 5: The higher the quality of government, the more significant the positive 167 relationship between the AIS and TFCEP. 

In lines 140-142 and 154 and next: 

“Because the spontaneous adjustment of the market often lags behind and is unstable, 140 AIS and GTI normally depend more on the government because it controls resources and 141 can intervene in resource allocation, so GQ is not only related to the effective implemen- 142 tation of GC and in lines 154-policies, but also determines the effect of GC on improving TFCEP.” 

“Effective government strengthens the impact of AIS on TFCEP through two ap- 154 proaches. First, it can avoid negative externalities of environmental pollution which will 155 lead to the failure of market mechanisms. Local governments adopt strict environmental 156 regulations, such as carbon tax and carbon quotas, to guide industrial enterprises to green 157 transformation and industrial structure upgrading. Secondly, the current international 158 political and economic environment is complex and changeable, and the negative impact 159 of the global financial crisis and public health emergencies on the global economy is ex- 160 tremely severe. An effective government can hedge against external shocks through pol- 161 icy regulations. Since the outbreak of COVID-19 in 2020, the Chinese government has im- 162 plemented some combined measures in the fields of finance, currency, and employment, 163 which not only maintains the bottom line of national life, health and safety, but also guides 164 industrial structure upgrading by vigorously developing the digital economy, and finally 165 leads the world economy in 2021.“

So why do you need hypotheses 2 and 3, if in hypotheses 4 and 5 you state that the relation exists? H4 assumes confirmation of the hypothesis H2, and H5 assumes confirmation of the hypothesis H3.

Author Response

(The authors gave the same response as above.)

Reviewer 4 Report

The paper is very interesting, just small revisions are needed. 

The first affirmation, lines 30-33, must be referenced. When does it happen?

TFCEP in line 41 is the first appearance in the text (not counting the abstract) than should be complete writen. ... "total factor..." 

"though the research on the relationship between finance and TFP is quite rich, few studies have examined the impact of GC on TFCEP." please quote some references to illustrate it.

What is the TFCEP, conceptually? who created it, and who proposed it? The main contribution is to use one indicator, so the authors must discuss deeply the indicator.

Also the Hypotheses H1 and H5, could be better referenced.

I recommended a graphic (figure) showing the hypotheses, and the effects, i.e. the direct impact, mediation, and moderation.   O suggest drawing it just after the Hypothesis 5.

What means lines 250-251? there's in no data from Hong Kong, Macau Taiwan and Tibet? 

The sources lines 252-254 must be included in the references. 

 The exclusions (441-442) must be discussed and maybe change. 

Which data are used? from which regions provinces (l.451-452)? Could the authors affirm that ?

Author Response

(The authors gave the same response as above.)

Round 2

Reviewer 1 Report

All comments have been answered.